# Analysis of Patch Antenna with Broadband Using Octagon Parasitic Patch

**DOI:** 10.3390/s21144908

**Published:** 2021-07-19

**Authors:** Sun-Woong Kim, Ho-Gyun Yu, Dong-You Choi

**Affiliations:** 1Team for Education and Research of Future ICT & AI Convergence, Chosun University, Gwangju 61452, Korea; woongskim1@naver.com; 2Department of Information and Communication Engineering, Chosun University, Gwangju 61452, Korea; 1030ghrbs@naver.com

**Keywords:** broad bandwidth, microstrip patch antenna, parasitic octagon patch

## Abstract

This paper proposes a novel broadband octagonal patch antenna with parasitic patches. The proposed patch antenna is constructed with four parasitic patches around a central radiating octagonal element. It is illustrated that this arrangement can be used to improve the antenna bandwidth and gain when compared with that of conventional antennas. The proposed patch antenna is very simple, low-profile, and economical. The typical analysis of the proposed antenna is analyzed by the S11(S-parameter), the radiation pattern, and the realized gain. It can achieve an impedance bandwidth of 1.44 GHz and a high gain of 8.56 dBi in the 8.5 GHz band. Furthermore, the proposed antenna shows that the directional pattern and HPBW measurement results of E and H-plane were 70° and 74° at 8.5 GHz, and 74° and 83° at 9 GHz, and 47° and 42° at 9.5 GHz, respectively.

## 1. Introduction

Recently, with high-speed technology and the rapid increase in demand, a microstrip patch antenna with compact structure, low profile, broad bandwidth, and easy integration is required [1]. However, the microstrip patch antenna must have a relatively narrow bandwidth compared to other antennas, and it is not suitable for application in systems that require a wide bandwidth [2,3].

A representative method for extending the bandwidth of the microstrip patch antenna is to chop part of the ground and increase the coupling induced between the antenna and the ground [4,5,6]. This method causes the loss of the directional radiation characteristic of these antennas, which is a unique characteristic. After that, they can be considered as monopole antennas with omnidirectional radiation characteristics. Therefore, the proposed antenna minimizes structural changes in the ground, maintains the directional radiation characteristics, and has a fractional bandwidth of 10% or more [7,8].

The bandwidth of the proposed antenna is extended through the four parasitic patches that exist independently on the top, bottom, left side, and right side of the central patch antenna. The coupling gaps between the main central and parasitic patches form the *E* and *H* fields, which can be interpreted as parallel *L* and *C* in an equivalent circuit. To verify these facts, we analyze the surface current distribution. The lower and central frequency bands form the current in the top and bottom parasitic patches in the proposed bandwidth. Finally, the upper-frequency band forms the current in the left-right parasitic patch. The current distribution on the main central patch and the parasitic patches contributes to a broad bandwidth of approximately 1.44 GHz.

To verify the practicality of the proposed antenna, we have compared similar antennas with advantages of the compact structure, broad bandwidth, high gain, and good directivity pattern, and it is presented in Table 1 [9,10,11,12,13]. Similar antennas have achieved broad bandwidth by forming multiple and single resonances through the parasitic patches. This paper also formed dual resonance modes through the four octagonal parasitic patches and achieved a broad bandwidth above 1 GHz.

The proposed antenna consists of four parasitic patches and a center patch. Due to the four octagonal parasitic patches, the dual resonance mode is formed in 8.61 to 9.14 GHz and 9.15 to 9.59 band. This phenomenon combines four parasitic patches, and the final antenna has a broad bandwidth of 8.47 to 9.62 GHz in dual resonance modes.

After comparing the proposed antenna with the existing literature, it is evident that the proposed antenna has some benefits considering performance or size, gain, and bandwidth. These privileges can be a good candidate for applications that require the communication of broad bandwidth.

## 2. Design and Analysis of the Proposed Antenna

The proposed antenna was designed using HFSS software (Ansys Co., Canonsburg, PA, USA), and it consists of four octagonal parasitic patches and a central octagonal patch. The antenna is fabricated as a laminated substrate structure using two Taconic RF-30 substrates that have a relative permittivity of 3.0, loss tangent of 0.0013, and thickness of 1.52 mm, and a rectangular slot is inserted into the ground at the center so that waves can be transmitted from the feeding structure in the rear to the radiation element in the front.

Based on the simulation results, the proposed fabricated antenna is illustrated in Figure 1 and Figure 2.

### 2.1. Comparison of Typical and Proposed Antennas

A typical microstrip patch antenna is simple and inexpensive to manufacture and easy to attach with the planar and non-plane spaces because it is thin. Furthermore, if used in printed circuit technologies, it is suitable for the combination of integrated circuits. However, it has a narrow band between 1 to 10% due to a high Q value. To overcome this challenge, we extended the bandwidth beyond 500 MHz through parasitic patches. The comparative analysis of the simulation and structure for each type of antenna are presented in Figure 3 and Figure 4.

The antenna’s impedance bandwidth is measured based on –10 dB, and the result of the simulation of the typical patch antenna is observed at a bandwidth of approximately 530 MHz, within 8.84–9.37 GHz. The proposed antenna is observed at a bandwidth of approximately 1.15 GHz, within 8.47 to 9.62 GHz. Therefore, the fractional bandwidths of the typical patch and proposed antennas are 5.82% and 12.71%, respectively, and the impedance bandwidth of the proposed antenna is extended by approximately two times.

The gains of the proposed and typical antennas have been analyzed, and the result has been illustrated in Figure 5.

As depicted in Figure 5, the gain for a typical antenna is only 7.15 dBi in 9 GHz, whereas it is 10.27 dBi in 9 GHz for the proposed antenna.

### 2.2. Comparison of Parasitic Patch

As illustrated in Figure 6, the coupling gap between the radiating element and the parasitic patch form guided the E and H fields, leading to the development of a broad bandwidth. The *E* and *H* fields guided between the radiating element and the parasitic patch are interpreted as parallel *L* and *C* in an equivalent circuit.

The resonance frequencies of the radiating element and parasitic patch follow the characteristics of *L* and *C*. The length of each element can be presented as the value of *L* in series, and as the length of the patch increases, the series *L* value increases. Furthermore, as the series *L* value increases, the total area of the patch increases. Thus, the *E* field induced between the patch and the ground increases [14,15,16]. Therefore, it is the shunt *C* from the high between the patch and ground plane. As illustrated in Figure 7, it can be confirmed that E-fields formed in the central patch and the parasitic patch. 

The design process of the proposed antenna is illustrated in Figure 8. The four octagonal parasitic patches of the proposed antenna were analyzed in several ways: top to bottom and left to right; a broad bandwidth was achieved through the parasitic patches.

The results of the simulation using the parasitic patches of a proposed antenna are presented in Figure 9.

The “Structure-1” of Figure 8, with the top-bottom parasitic patches, generates resonance at lower frequencies of 8.62–9.14 GHz. Additionally, the “Structure-2” of Figure 8, with the left-right side patches, generates resonance at higher frequencies of 9.15–9.59 GHz. The final structure generates resonance at the lower and higher frequencies in the 8.47–9.62 GHz band.

The simulated surface current distribution at 8.5, 9, and 9.5 GHz is presented in Figure 10, which shows the resonance formations for each patch. 

As illustrated in Figure 10, the current is formed at the lower frequency of the 8.5 GHz band in the top-bottom parasitic patch around the central patch. Likewise, the current is formed at the central frequency of the 9 GHz band and the higher frequency of the 9.5 GHz band in the top-bottom and left-right parasitic patches, respectively.

### 2.3. Analysis for Dimension Variation of Proposed Antenna

The dimension (radius *r*) of the proposed antenna is a significant variable for generating resonance in the desired band, and the analysis of the simulation for the variation in radius *r* is presented in Figure 11.

As presented in Figure 11, when the radius *r* of the patch takes the values 3.61, 5.61, and 6.61 mm, a low-resonance point and narrow bandwidth are observed. However, when the radius *r* = 4.61 mm, a maximum resonance of –22.10 dB was observed in the 8.92 GHz band, and the impedance bandwidth was approximately 1.15 GHz, within 8.47–9.62.

The simulation analyses for distance, d, at the central patch and the parasitic patches are presented in Figure 12.

As indicated in Figure 12, the bandwidth results were 0.6 GHz within 8.45–9.05 GHz for *d* (distance variation) = 0.16 mm, 1.15 GHz within 8.47–9.62 GHz for *d* = 0.26 mm, 0.98 GHz within 8.59–9.57 GHz for *d* = 0.36 mm, 0.85 GHz within 8.66–9.51 GHz for *d* = 0.46 mm, and 0.72 GHz bandwidth within 8.73–9.45 GHz for *d* = 0.56 mm. Thus, for *d* = 0.26 mm and *r* = 4.61 mm, the widest impedance bandwidth was observed and finally selected.

## 3. Experimental Results

From the results of the simulation of the impedance bandwidth of the proposed antenna presented in Figure 13, a broad bandwidth of approximately 1.15 GHz within 8.47–9.62 GHz was observed based on –10 dB. Furthermore, the measurement of the impedance bandwidth for the fabricated antenna yielded a broad bandwidth of approximately 1.44 GHz within 8.17–9.61 GHz.

The radiation pattern results of the fabricated antenna were presented in Figure 14 and Figure 15. 

As depicted in Figure 14 and Figure 15, the directivity patterns of the measured and simulated radiation for the proposed antenna indicate that the antenna gain is concentrated at 0°. In the case of simulation and measurement results for the HPBW (Half Power Beam Width), the E-plane(xz-plane) were 52° and 50° at 8.5 GHz, 40° and 59° at 9 GHz, and 88° and 88° at 9.5 GHz. Furthermore, the H-plane(yz-plane) results were 70° and 74° at 8.5 GHz, 74° and 83° at 9 GHz, and 47° and 42° at 9.5 GHz, respectively.

The antenna gain analyses of the proposed antenna within 8.5, 9, and 9.5 GHz bands are presented in Figure 16.

The results of simulated gain of the proposed antenna are 9.2, 10.2, and 8.9 dBi in the 8.5, 9, and 9.5 GHz bands, respectively, and the results of the measurement are 8.56, 7.83, and 8.92 dBi in the 8.5, 7.83, and 9.5 GHz bands, respectively.

Between the simulation and measurement results, a slight difference is noticed. There are chances to develop this discrepancy which can be explained by one or both of the following reasons. First, since the antenna was made using the etching technique, there may be a blunder in the fabrication process. The etching process can cause a shift in the measurement dimension of several millimeters. Second, there may be a loss between the antenna and the measuring cable that has not been taken into account. However, this slight discrepancy between simulation and measurement results does not prevent the antenna from being used in practical applications. 

In future work, the fabricated antenna should be applied at the location tracking system of the non-contact type. A candidate technology for the system is the IR-UWB radar module of Novlda’s NVA-R661, and an X2 chip used in the NVA-R661 module has 0 to 10 channels. Among them, the center frequency of channel 10 is an 8.8 GHz band, which is identical to the center frequency of the fabricated antenna. Therefore, it is shown that the proposed antenna is suitable for the use of the NVA-R661 module.

## 4. Conclusions

An octagonal antenna with four parasitic patches is proposed in this paper. We have illustrated that by incorporating a central octagonal patch and four parasitic patches, both the impedance bandwidth and the antenna gain can be improved. We confirmed that when placing two parasitic patches at the top-bottom around the central patch, the primary resonance is formed. Furthermore, the second resonance is formed when two parasitic patches are placed at the left-right around the central patch. This phenomenon, when combined with each patch, can achieve a broad bandwidth above 1 GHz. 

The proposed antenna has the advantages of low-profile, broad bandwidth, high gain, and good directivity, and this advantage would be a good candidate for applications that require the communication of broad bandwidth.

## Figures and Tables

**Figure 1 sensors-21-04908-f001:**
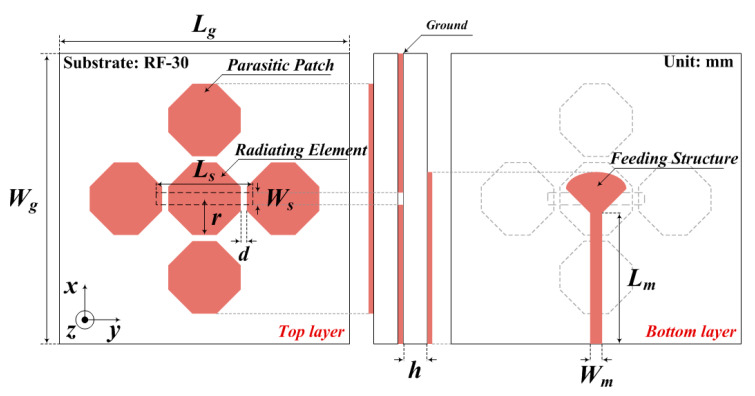
Structure and dimensions (in mm) of the proposed antenna: *L_g_* = 50, *W_g_* = 50, *L_s_* = 1.2, *W_s_* = 0.9, *L_m_* = 25.45, *W_m_* = 4.6, *r* = 4.61, and *d* = 0.26.

**Figure 2 sensors-21-04908-f002:**
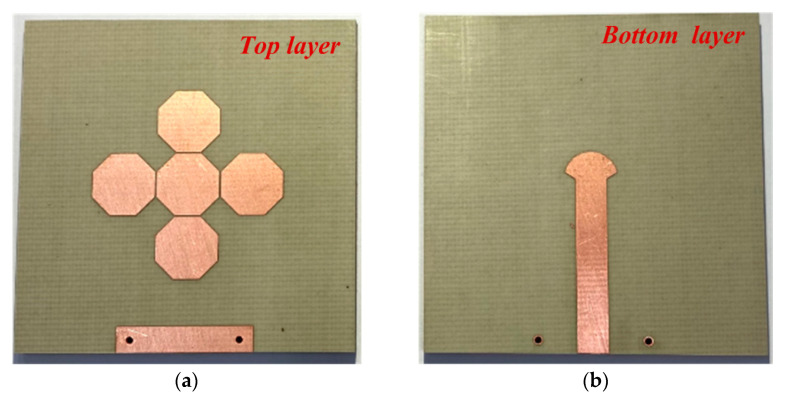
Photographs of the fabricated antenna: (**a**) top layer, (**b**) bottom layer.

**Figure 3 sensors-21-04908-f003:**
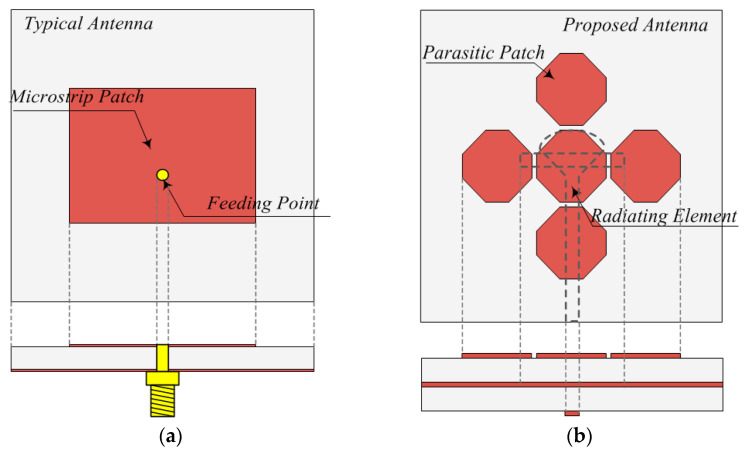
Structures of a typical patch and proposed antennas. (**a**) typical patch antenna, (**b**) proposed antenna.

**Figure 4 sensors-21-04908-f004:**
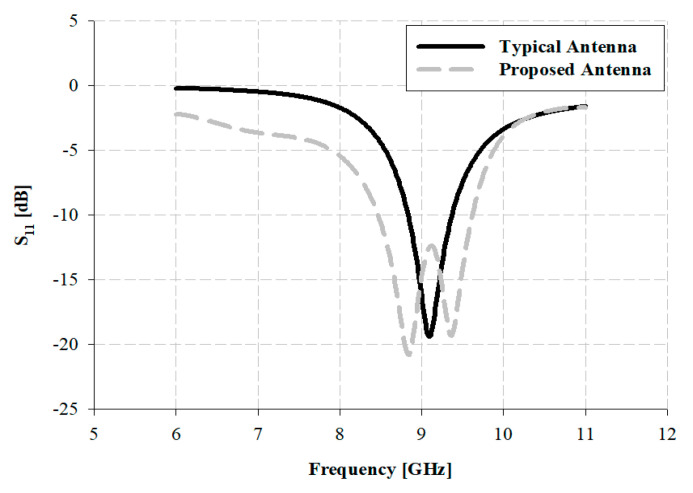
Analysis of comparison of the simulation of impedance bandwidth for each antenna.

**Figure 5 sensors-21-04908-f005:**
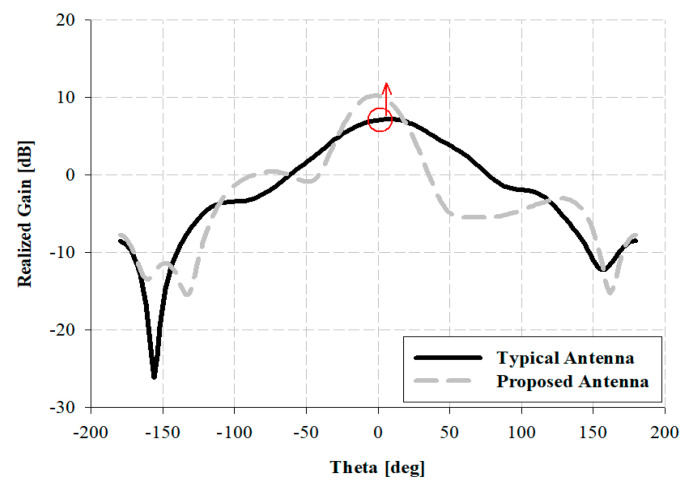
Simulation comparison analysis of radiation pattern for each antenna.

**Figure 6 sensors-21-04908-f006:**
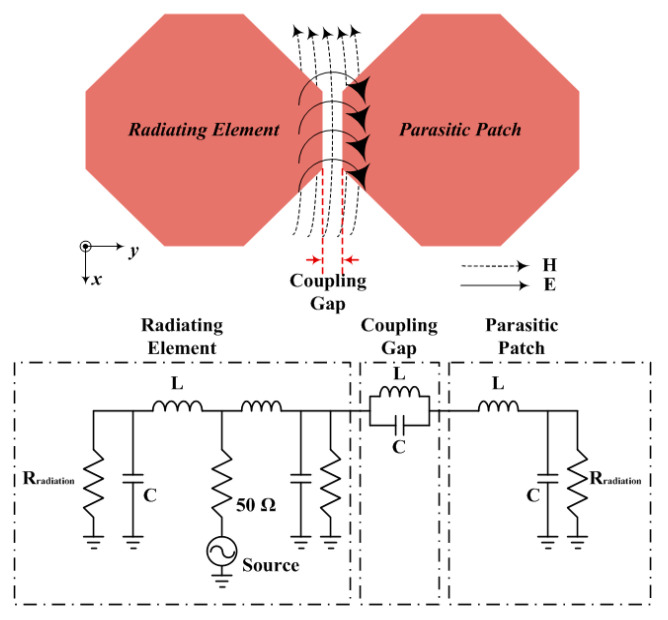
Equivalent circuit between the radiating element and parasitic patch.

**Figure 7 sensors-21-04908-f007:**
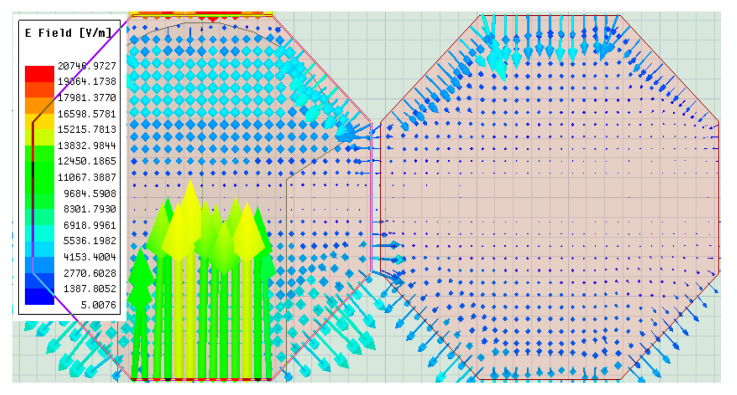
Analysis of current distribution between the radiating element and parasitic patch.

**Figure 8 sensors-21-04908-f008:**
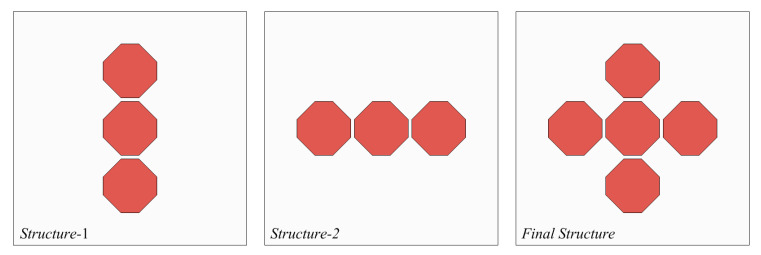
Design process of the proposed antenna.

**Figure 9 sensors-21-04908-f009:**
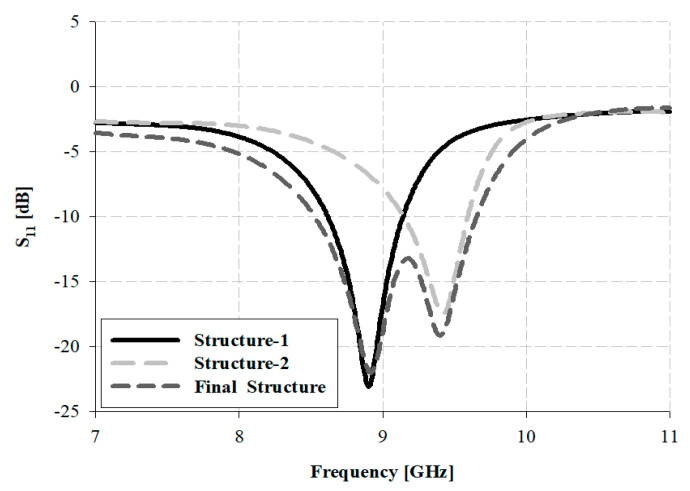
Results of simulation for the design process.

**Figure 10 sensors-21-04908-f010:**
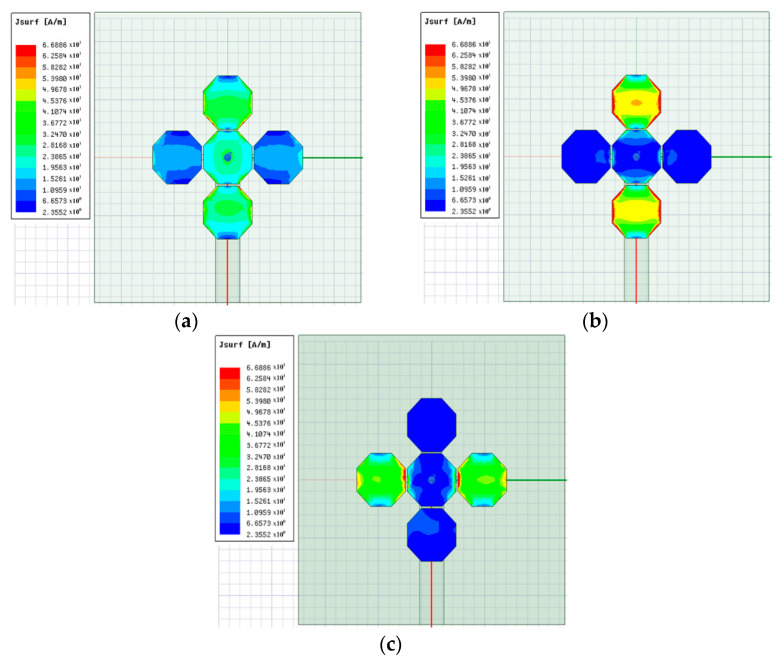
Analysis of the simulation of the surface current distribution of the proposed antenna: (**a**) 8.5 GHz, (**b**) 9 GHz, (**c**) 9.5 GHz.

**Figure 11 sensors-21-04908-f011:**
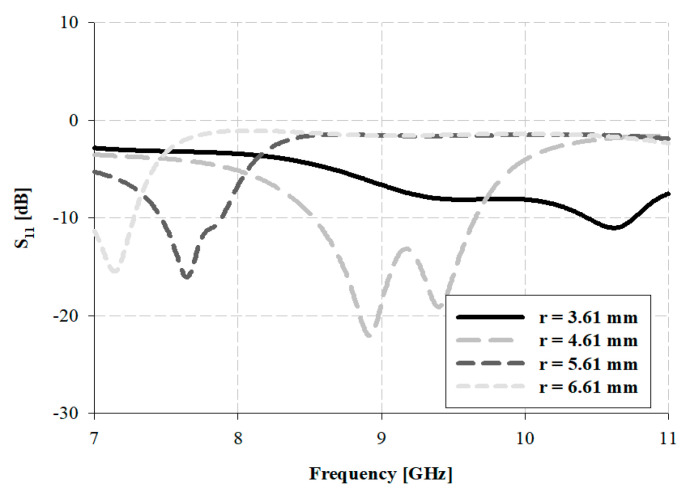
Analysis of the simulation with variation in r for the proposed antenna.

**Figure 12 sensors-21-04908-f012:**
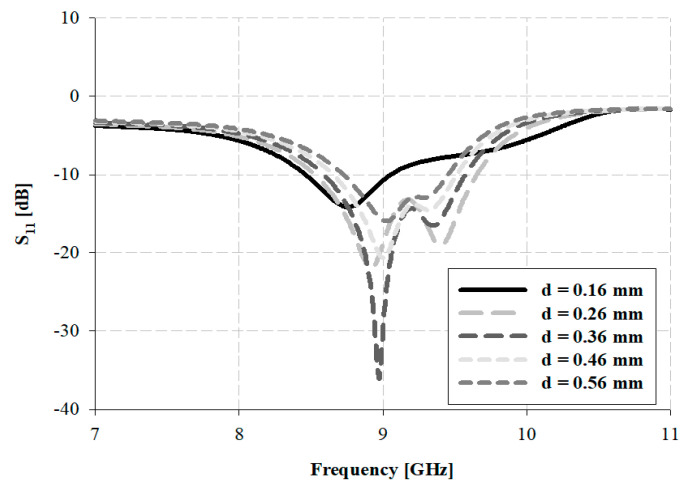
Analysis of the simulation for distance d of the proposed antenna.

**Figure 13 sensors-21-04908-f013:**
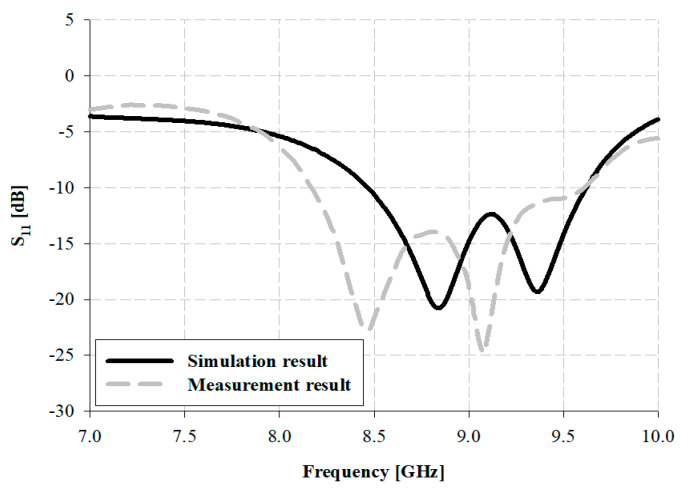
Simulation of impedance bandwidth and results of measurement for the fabricated antenna.

**Figure 14 sensors-21-04908-f014:**
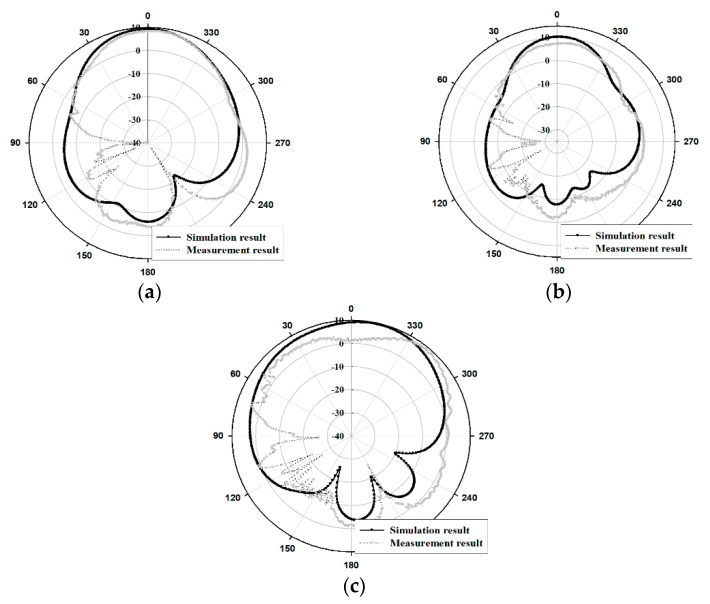
Radiation pattern results of the simulation and the measurement in E-plane(xz-plane): (**a**) 8.5 GHz, (**b**) 9 GHz, (**c**) 9.5 GHz.

**Figure 15 sensors-21-04908-f015:**
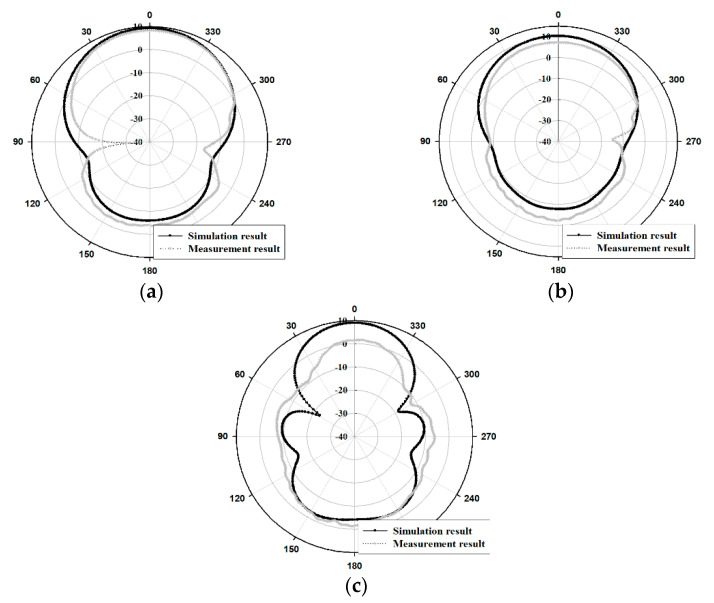
Radiation pattern results of the simulation and the measurement in H-plane(yz-plane): (**a**) 8.5 GHz, (**b**) 9 GHz, (**c**) 9.5 GHz.

**Figure 16 sensors-21-04908-f016:**
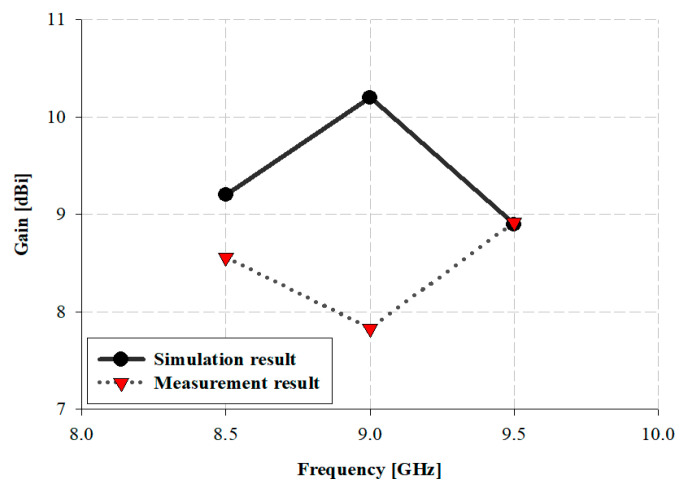
Results of the measurement of the proposed antenna: (**a**) 8.5 GHz, (**b**) 9 GHz, (**c**) 9.5 GHz.

**Table 1 sensors-21-04908-t001:** Comparison of the proposed antenna and reference antennas.

Ref.	Antenna Type	Size (mm^2^)	Resonant Band (S11 ≤ −10 dB) (GHz)	Band Width (GHz)	Fractional Bandwidth (%)	Gain (dBi)	Radiation Pattern
2018 [9]	Patch with multiple open slot	57 × 72	1.77–2.65	0.88	39.81	8.5	directivity
2018 [10]	Patch with parasitic	76.3 × 63.1	5.03–5.28	0.25	4.84	12.15	directivity
2018 [11]	Array Patch with parasitic	-	3.35–3.95	0.6	16.43	13.6	directivity
2018 [12]	Patch with parasitic	36 × 39	5.46–6.27	0.81	13.81	-	directivity
2020 [13]	Patch with parasitic	56 × 52	2.09–2.64	0.55	23.25	4.54	directivity
This work	Patch with parasitic	50 × 50	8.17–9.61	1.44	16.19	8.92	directivity

## Data Availability

Not applicable.

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
