# Peer review of "Analysis of Patch Antenna with Broadband Using Octagon Parasitic Patch"

_sensors, 2021, doi:10.3390/s21144908_

Round 1
Reviewer 1 Report
Very interesting and relevant report on broad band patch antenna. The text and quality of the document got to be improved tough.
-"Furthermore, the proposed antenna shows the directional pattern in which the wave reaches a specific direction" -- should omit the last part about reaching a specific direction. Besides the gain/directivity should be mentioned.
-"The coupling gaps between the main central and parasitic patches form the E and H fields," -- Gaps do not "form" E and H fields, this sentence should be rewritten.
-"Similar antennas are 41 achieved broad bandwidth" -- have achieved
-Table I contains the bandwidth in absolute terms (GHz), it should present the relative (%) figure instead, since it is not clear all other elements operate in the same frequency range.
-"plane and nonplane" -- use instead planar and non-planar
-"it has a narrow band of dozens of megahertz orders" - wrong, authors should use relative % values instead.
- Fig 3 has the legend presented backward.
- Authors point out fabrication problems as the cause for the difference between simulation and measurement. From fig. 13, it seems that the shift towards smaller frequencies, while keeping the same S11 shape can be ascribed to a different eps_r.
Reviewer 2 Report
I feel very conflicted about this paper. On one hand the idea behind the article is interesting and could be published. On the other hand, the presentation of the paper is not on point. The real novelty/improvement of the work is obscure. And there are some other issues that concern me.
First of all, you should meticulously check your paper for English grammar and style. Some sentence make no sense at all and many others are hard to read. Just to name a few examples: row 39-40, 45-48, 55-57. If you encounter difficulties, please ask a native English speaker for advice, or someone that is fluent.
In table 1, the comparison does not give any insight on the improvements over the scientific literature that your antenna should bring. According to this table, the proposed antenna does not retain the smallest size, nor the higher gain. Maybe is a combination of size, gain and bandwidth? However, how can I evaluate the bandwidth if you present me an absolute value? You should provide fractional bandwidths, and also, for the sake of completeness, the operating frequency of the cited works.
Figure 3a is flipped, please adjust.
Row 93: "the impedance bandwidth of the proposed antenna is extended by approximately three times". No, it is not! It is about two times.
Figure 13: Simulations and measurements do not agree very well. I understand that you provided some explanations for this behavior, but I do not think that this is acceptable. You should repeat your measurements paying attention to the measurement process. If the problem is on the precision of the etching process you should consider using a company that can do the work for you. If this solution is not feasible, at least provide a parametric simulation of the S11 taking into account the error that your etching process may cause.
Round 2
Reviewer 2 Report
I have just one last request. You should add the information you provided within the author's reply about the employment of the antenna inside the NVA-R661 module. This would at least partially mitigate the discrepancy between measurement and simulations.
